# miRNAs in Neurological Manifestation in Patients Co-Infected with SARS-CoV-2 and Herpesvírus 6 (HHV-6)

**DOI:** 10.3390/ijms241311201

**Published:** 2023-07-07

**Authors:** Vanessa Cristine de Souza Carneiro, Otacilio da Cruz Moreira, Wagner Luis da Costa Nunes Pimentel Coelho, Beatriz Chan Rio, Dmitry José de Santana Sarmento, Andreza Lemos Salvio, Soniza Vieira Alves-Leon, Vanessa Salete de Paula, Luciane Almeida Amado Leon

**Affiliations:** 1Laboratory of Molecular Virology and Parasitology, Oswaldo Cruz Institute, Fiocruz, Rio de Janeiro 21040-360, Brazil; 2Laboratory of Technological Development in Virology, Oswaldo Cruz Institute, Fiocruz, Rio de Janeiro 21040-360, Brazill_amado@ioc.fiocruz.br (L.A.A.L.); 3Real Time PCR Platform RPT09A, Fiocruz, Rio de Janeiro 21040-360, Brazil; 4Department of Oral Diagnosis, School of Dentistry, State University of Paraíba, Araruna 58429-500, Brazil; 5Laboratory of Translacional Neurosciences, Biomedical Institute, Federal University of the State of Rio de Janeiro-UNIRIO, Rio de Janeiro 22290-240, Brazil; 6Department of Neurology, Reference and Research Center for Multiple Sclerosis and Other Central Nervous System Idiopathic Demyelinating Inflammatory Diseases, Clementino Fraga Filho University Hospital, Federal University of Rio de Janeiro, Rio de Janeiro 21941-617, Brazil

**Keywords:** microRNA, human herpesviruses 6, neurological manifestation

## Abstract

Human herpesviruses (HHVs) can establish latency and be reactivated, also are neurotropic viruses that can trigger neurological disorders. HHV-6 is a herpesvirus that is associated with neurological disorders. Studies have reported the detection of HHV-6 in patients with COVID-19 and neurological manifestations. However, specific diagnoses of the neurological disorders caused by these viruses tend to be invasive or difficult to interpret. This study aimed to establish a relationship between miRNA and neurological manifestations in patients co-infected with COVID-19 and HHV-6 and evaluate miRNAs as potential biomarkers. Serum samples from COVID-19 patients in the three cohorts were analyzed. miRNA analysis by real-time polymerase chain reaction (qPCR) revealed miRNAs associated with neuroinflammation were highly expressed in patients with neurological disorders and HHV-6 detection. When compared with the group of patients without detection of HHVs DNA and without neurological alterations, the group with detection of HHV-6 DNA and neurological alteration, displayed significant differences in the expression of mir-21, mir-146a, miR-155 and miR-let-7b (*p* < 0.01). Our results reinforce the involvement of miRNAs in neurological disorders and provide insights into their use as biomarkers for neurological disorders triggered by HHV-6. Furthermore, understanding the expression of miRNAs may contribute to therapeutic strategies.

## 1. Introduction

Herpesviruses are DNA viruses that belong to the herpesviridae family. These viruses can be divided into 3 subfamilies: Alphaherpesvirinae, Betaherpesvirinae, and Gammaherpesvirinae [1]. There are 9 types of herpesvirus that can infect humans, and all can establish latency and be reactivated upon changes in host immunity [2]. In addition to the classic symptomatology, replication of the herpesvirus can be associated with several diseases, such as cancer and neurological diseases [3,4,5].

Herpesviruses are known to be neurotropic [6]. The Betaherpesvirinae, Herpesvirus Human 6 (HHV-6), is a herpesvirus that has been extensively studied in recent years to derive its association with neurological outcomes, such as encephalitis, meningitis, seizures, headache, and epilepsy, and its involvement in neurodegenerative diseases [7,8,9].

Infection with the SARS-CoV-2, which causes COVID-19, is known to trigger a state of immunosuppression in some patients, which may cause the reactivation of herpesviruses [10]. Although SARS-CoV-2 has been detected in the central nervous system (CNS), its mechanisms of action in the nervous system remain unclear [11]. According to case reports, patients with COVID-19 develop neurological alterations caused by the reactivation of HHV-6 and not by the action of SARS-CoV-2. Moreover, herpervirus encephalitis can evaluated to autoimmune encephalitis associated to antibodies to the receptor of N-methyl-D-Aspartate (NMDA) [12], and recently registers showed that COVID-19 infection can also be followed by autoimmune encephalitis mediated by antibodies to glycoprotein (Ab-GAD) or to MOGAD disease [13,14,15]. Thus, the action of this herpesvirus should be investigated in the context of COVID-19 [16,17] not only directly acting in the nervous system, but also as trigger to the new manifestations in specific individuals. Recently, we analyzed nasopharyngeal swab samples collected from hospitalized patients with COVID-19 confirmed, to investigate the frequency of the detection of all types of herpesviruses in critically ill patients with COVID-19, and thus, to evaluate if SARS-CoV-2 infection could be a risk factor for Herpesviridae reactivation. It was observed that CNS symptoms were more prevalent in patients with HHV-6 detection (40% in patients with HHV-6 detection vs. 14.3% in patients without HHV-6 detection), showing a statistically significant association between HHV-6 detection and neurological symptoms in patients with COVID-19 [18]. The detection of CMV, HHV-6 and HHV-7 from nasal swab has been used to investigate Herpesviridae reactivation in previous studies [19,20], since the detection of herpes close to the sites of replication may indicate reactivation.

Virus DNA or RNA can be detected via nucleic acid amplification tests (such as PCR) of cerebrospinal fluid (CSF) samples to identify etiological agents in patients with neurological disorders, such as encephalitis [21]. Accordingly, studies have sought to detect HHV-6 DNA in the CNS to derive the association between neurological alterations and HHV-6 infection [17,22,23]. However, conflicting views exist regarding the detection of HHV-6 DNA in the CNS as a diagnostic tool for neurological diseases caused by HHV-6 [9,24]. In addition, this method is associated with difficulties in the collection of CSF [25]. Currently, one of the main challenges faced by clinicians is the lack of reliable diagnostic tools that can enable the diagnosis of neurologic alterations or neurologic inflammation as early as possible. An ideal biomarker must have high sensitivity and specificity, easy to measure, minimally-invasive to collect, provide reproducible results, and accurately reflect disease progression. Molecular biomarkers, including microRNAs (miRNAs), may be measured in two extracellular fluids that are of the highest importance in neurological alterations, cerebrospinal fluid (CSF) and blood. miRNAs are small RNAs comprising approximately 22 nucleotides that act as post-transcriptional regulators by binding in a partially complementary manner to the target mRNA, causing mRNA degradation or silencing [26]. Thus, miRNAs are involved in several cellular processes, such as the regulation of proteins involved in immune pathways [27], and can be measured in body fluids using techniques, such as qPCR [28]. As a result, miRNA profiles have been proposed as biomarkers of various pathologies, including neurological diseases. Further, serum collection is less invasive and easier to handle, and contains miRNA from all tissues in the body [29]. Notably, miRNAs that circulate in serum are relevant biomarkers of diseases of the central nervous system, controlling the initiation and maintenance of inflammation. miR-155 is one of the main investigated miRNAs in the context of neuroinflammation and is considered one of the most potent pro-inflammatory miRNAs [30,31], miR-155 induces neuroinflammation by inhibiting factors involved in the inflammatory process, some targets of miR-155 include anti-inflammatory regulators such as Suppressor of cytokine signaling 1 (SOCS1). Cardoso et al., 2012 [32] showed that miR-155 expression would be associated with inflammation-mediated neuronal cell death. Therefore, this miRNA has been observed in several other neuropathologies. MiR-21 and miR-146a are also constantly associated with neuroinflammation, are considered anti-neuroinflammatory miRNAs [33,34,35]. miR-21 down-regulates the inflammation process, acting to maintain homeostasis by turning off the excessive, often harmful, “pro-inflammatory” response [33] miR-146a, is another miRNA constantly associated with a suppressor function in neuroinflammation, reducing the expression of pro-inflammatory cytokines [34]. Despite being a negative regulator of inflammation, miR-146a is up-regulated in the pathogenesis of several neurological conditions, suggesting that cells are compensating for pathological inflammation and attempting to restore homeostasis [36], miRNAs from the let-7 family can act as both pro-inflammatory and anti-inflammatory, and miR-let-7b is associated with neuroinflammatory stimuli [36,37]. Some studies have shown that the toll like receptor 7 (TLR7) expressed in microglia can be activated by miR-let-7b, promoting the activation of immune cells in the CNS and neuroinflammation [37].

Many studies have described the potential of miRNAs as biomarkers of the severity and outcome of COVID-19 [38,39]. In addition, miRNA profiling has also been investigated as biomarkers for neurological disease [40,41,42]. This study aimed to establish a relationship between miRNA and neurological manifestations in patients with COVID-19 and HHV-6, and to evaluate miRNAs as potential biomarkers of this pathogenesis.

## 2. Results

### The Expression of miRNAs Associated with Neuroinflammation in Patients with COVID-19

We individually analyzed the expression of selected miRNAs among the three COVID-19 patient groups to explore the possible association of these miRNAs with neuroinflammation and HHV-6 action.

The main demographic and clinical data of patients are summarized in Table 1. The mean patient age ranged from 59 to 71 years, with hypertension as the most prevalent comorbidity. Group 2 had worse outcomes, such as a Sequential Organ Failure Assessment SOFA score > 9 (60%) and death (60%), than the other groups (Table 1).

Patients in Groups 1 and 2 had neurological alterations, as shown in Table 2. The predominant symptoms were impaired consciousness and headaches. Some patients in Group 2 had symptoms associated with the peripheral nervous system.

A significant difference in the expression of all microRNAs (miR-21, miR-146a, miR-let-7b, and mi-155-5p) could be confirmed among the validation groups. The expression levels of miR-21, miR-146a, and miR-let-7b were higher in Group 1 (FC 7.19; 11.38; 50.12) than in Group 2 (FC 1.47, 4.03, 8.36) and 3 (FC 1.51; 1.22; 2.39), while, miR-155 was higher in Group 2 than in Groups 1 and 3. The expression of miR-155 was significantly increased in Groups 1 and 2 (FC 24.70 and 30.01) compared to those in Group 3 (FC 1.08). For mir-21 there was a significant increase in expression in Group 1 compared to Group 2 (*p* = 0.043). A significant increase of the levels of all miRNAs was seen in Group 1, compared to those in Group 3, miR-21 (*p* = 0.007); miR-155 (*p* = 0.001); miR-146a (*p* = 0.002); and miR-let-7b (*p* = 0.005) (Figure 1).

## 3. Discussion

In the present study, we evaluated the expression of miRNAs associated with neuroinflammation in patients with COVID-19, neurological symptoms, and HHV-6, a neurotropic virus. Some cases reported of neurological manifestations in patients with COVID-19 were triggered by HHV-6 reactivation [16,17]. But studies have also indicated that HHV-6 is a viral contributor to the development and progression of long COVID, a multifactorial disease [43].

In our assessment a significantly altered level of neurological manifestation-associated miRNAs was identified in COVID-19 patients, highlighting the association of COVID-19 with neurological diseases and respective biomarkers. This association was more evident with miR-155, which significantly differed in the two groups with neurological alterations compared to Group 3 without neurological alterations and with COVID-19. Such finding suggests that miR-155 can be studied as a neuroinflammation biomarker. In an analysis of neuroinflammatory biomarkers, Keikha et al. [35] revealed a significant increase in relative expression in patients with severe COVID-19. Although miR-155 is widely associated with inflammation and is well-described to be increased in patients with COVID-19 [44], our findings suggest that the increase in miR-155 expression in patients with COVID-19 may be associated with its pro-neuroinflammatory function. miR-155-5p is a pro-inflammatory miRNA that is commonly upregulated in neurological disorders. Prior studies have also revealed miR-155 as a biomarker in the pathogenesis of ischemic stroke [45,46]. According to surveys, an increase in this miRNA may be associated with brain damage observed in diseases, such as multiple sclerosis (MS) [36]. Furthermore, MiR-155 is also investigated in the regulation of neuroinflammation induced by viral infections. Azouz et al., 2019 [47] in their research with neurons from mice infected with the Zika virus (ZIKV), a virus known to cause a wide spectrum of neurological diseases, was observed that miR-155 had an increase in its expression. The same profile evidenced by Majer et al., 2017 for HSV-1 [48].

The expression levels of miR-21, miR-146a, and miR-let-7b were higher in patients with neurological disorders and co-infection with HHV-6 than in patients without HHV-6. Neurological alterations were identified in Group 1, with detection of HHV-6, including headache, impaired consciousness, and acute cerebrovascular disease. Patients in Group 2, in which HHV-6 was not detected, experienced headache, impaired consciousness, ageusia, anosmia, and vision impairment. miR-21-5p has been suggested to be a biomarker of neurological diseases because it is closely associated with the regulation of almost all major CNS disorders, including ischemic stroke, neurodegenerative diseases, CNS tumors, epilepsy, trigger to autoimmune CNS diseases and CNS trauma [49].The main function of this miRNA is to regulate the apoptosis and activation of inflammatory processes in the nervous system. A previous study revealed the greater expression of miR-21 in patients with epilepsy [50]. The same was evidenced by Zhou and Zhang 2014 [51]. Who observed upregulated levels of miR-21 in patients with ischemic stroke (IS), indicating that high levels of this miRNA can be observed in different neurological manifestations. In our study we showed that mir-21 had significantly increased expression in Group 1 (patients with COVID-19, neurological alterations, and detection of HHV-6) compared to Groups 2 and 3. However, to our knowledge, this is the first study to demonstrate an association between this expression and HHV-6 co-infection. Evidence relate HHV-6 to various neurological diseases; reactivation of this virus could cause neurological symptoms. Notably, the detection of HHV-6 was associated with delirium, cognitive decline, headache, and other symptoms reported in our study cohort [52,53].

In the present study, the expression of miR-146a was observed to significantly differ among the patient Group 1 (patients with COVID-19, neurological alterations, and detection of HHV-6) and Group 3 (patients with COVID-19). miR-146a is reported to be one of the main miRNAs that regulate neuroinflammation, usually with a suppressive effect [54]. Studies have linked the differential expression of mir-146a to neurological disorders, such as Alzheimer’s disease [55]. Although no studies have revealed an association between neurological alterations caused by HHV-6 and miR-146a, in neuronal cells infected with HSV-1, miR-146a has been found to be up-regulated [56]. Increased expression of miR-146a leads to the negative regulation of factor H in the complement system, which is responsible for antiviral defense [56]. In an in vivo study on HSV-1 encephalitis in SJL/J mice, Majer et al. [48] observed increased expression of miR-146a and miR-155. Taken together, our results suggest that these miRNAs should be investigated in the context of neurological manifestations and are promising biomarkers.

miRNAs of the let-7 family are among the most abundant miRNAs in the brain. Let-7 activates Toll-like receptor 7, which contributes to the spread of damage to the CNS. Individuals with Alzheimer’s disease have increased levels of let-7b in their cerebrospinal fluid (CSF) [37,57] miR-let-7b expression was found to be significantly increased in Group 1 compared to Group 3, despite a lack of reports on the association of miR-let-7b with HHV-6, is associated with neurological disorders, such as ischemic stroke [58]. However, miR-let-7b has been proposed as a biomarker of anti-NMDA receptor encephalitis, and as already mentioned anti-NMDA receptor encephalitis can be induced by herpesvirus whit HSV [57,59]. These findings encourage for future investigations that focus on the role of this miRNA during neurological complications triggered by herpesvirus, aiming at its use as a biomarker or in therapeutic treatment.

In this study, we sought to investigate association between the expression of these miRNAs and the detection of HHV-6 in COVID-19 patients with neurological symptoms, which was observed in Group 1. Due to the low incidence of HHV-6 detection in patients without neurological symptom (14.3%) [18], it wasn’t included a control group of neurological symptoms (−) and HHV-6 (+) in this study. Therefore, it is unclear whether the increased expression of miR-21, miR146a and miR-let-7b in Group 1 is related to neurological symptoms with HHV-6 (+) or HHV-6 only (+). However, we observed that miR-21, 146a and let-7 did not show significant difference between Groups 2 and 3. Moreover, CNS disorders tended to be higher in Group 1 than Group 2. As described above, the association of these miRNA with CNS disorders has already been demonstrated in literatures. The low incidence of HHV-6 detection in COVID-19 patients without neurological symptom in our previous study, it strongly suggests that HHV-6 reactivation induces CNS disorders and was the main cause of these miRNAs elevations, corroborating neurological manifestations during COVID-19. The expression level of all miRNAs was more elevated among Group 2, compared with Group 3, except for miRNA 21, that showed a slightly increase among Group 3. However, these differences were not statistically different. These findings reinforce the role of these miRNAs in neurological symptoms.

The expression of this miRNAs should be investigated in neurological disorders caused by HHV-6 to establish a relationship between neurological alterations and HHV-6 infection; this is because miRNAs are present in body fluids, such as blood, and can represent an alternative to CSF collection, which is an invasive technique and presents risks [29].

Neurological alterations caused by HHV-6 are constantly reported, and the symptoms presented in our study cohort are described during HHV-6 infection [60]. This opportunistic virus, when reactivated, triggers neurological manifestations in patients with COVID-19. The lack of an adequate diagnosis can impact the prognosis of the disease. The correct diagnosis provides fast recovery and absence of sequela, as described in the literature by Di Nora et al. [16] that reported a case of subsequent seizures in which the patient tested positive for HHV-6 and SARS-CoV-2, but had complete improvement with specific treatment. Another case was reported by Jumah et al., 2021 [17] who reported a case of patient with COVID-19, which developed myelitis due to HHV6. The patient showed complete neurological recovery with ganciclovir. Specific antiviral treatment for herpesviruses led to clinical improvement in the patient, indicating that an accurate diagnosis enables specific treatment and better patient prognosis. Therefore, it is important to investigate the potential biomarkers that enable correct diagnosis [16,17].

Understanding the role of miRNAs in neurological manifestations can also be useful to clarify cases of long COVID, where Patients who have had COVID-19 have reported cognitive deficits, headaches, and other neurological symptoms. Studies has been linked this condition to the reactivation of latent viruses such as HHV-6 and EBV [43,61].

In this study, we were the concerned about distinguishing the groups, avoiding confounding factors, which is common in studies that evaluate the expression of miRNAs. Thus, in Group 2, it was included patients who did not detect any HHVs. Furthermore, the selection of samples was made evaluating several intrinsic factors, such as homogeneity of pair samples between groups, considering age and detection of HHVs. This study had some limitations, including a small sample size and the inclusion of patients with comorbidities. However, given this specific cohort and the many variables to be analyzed, it is difficult to establish a homogeneity criterion for all aspects.

Notably, other studies that analyzed the expression of miRNAs in patients with COVID-19 outlined the same strategy, given the clinical characteristics of hospitalized patients [38].

Determining miRNA expression in serum represents a major challenge. In our study, we adopted some promising strategies employed by other groups that also work with the analysis of serum miRNAs. For example, as the amount of RNA in the serum is low and cannot be accurately determined for individual assays, the amount of RNA was based on a consistent input volume for all samples, as previously described [38]. As there is no consensus in the literature on the circulating miRNAs to be used as good reference miRNAs, we opted to use an synthetic exogenous miRNA as reference, cel-miR-39, which has been proposed in other studies [61].

In conclusion, all miRNA targets in the different groups displayed significant differences in expression and were more highly expressed in the group of patients with HHV-6. Therefore, miRNAs that participate in the regulation of neuroinflammation may be promising biomarkers for neurological disorders caused by HHV-6. Accordingly, HHV-6 infection should be assessed in patients with COVID-19 and neurological symptoms, including in the context of symptoms related to long COVID, such as headache, cognitive impairment, mental disorders, among others. The detection of HHV-6 is also associated with this condition; however, more studies are needed to clarify the role of herpesviruses in long COVID-19. Investigating the expression of miRNAs goes beyond establishing biomarkers. Alterations in the miRNA profile, such as those observed in our study, may be involved in the pathogenesis of neurological diseases and, therefore, may be potential therapeutic targets [62]. One example is miR-155, investigated in our study, which has already been proposed as a promising target to control neuroinflammation in AD [63]. It is evident that investigating the expression of miRNAs and their association with neurological disorders is an intelligent bet for promising therapies in the future.

## 4. Materials and Methods

### 4.1. Patients and Samples

In this study it was included serum samples from a cohort of 53 patients admitted to the Clementino Fraga Filho University Hospital (HUCFF) in Rio de Janeiro City, Brazil, with a molecular diagnosis of SARS-CoV-2 positivity by RT-PCR [29]. In our previous study, patients with mild or severe COVID-19 were further categorized according to their HHV-6 infection status and neurological diseases. 26.4% of them had neurological symptoms associated with the central nervous system, and among them 40% had HHV-6 DNA detected [18]. For the present study, it was selected 15 patients of these cohort, after matching for comorbidities, sex, and age group. Samples were collected after the onset of respiratory symptoms, and any change in the neurological status of patients after collection was observed and evaluated, by neurologists who collaborated in the study. However, the time between the onset of respiratory symptoms and sample collection may vary between patients. All patients were followed-up with a neurological evaluation at the time of discharge. Two trained investigators reviewed the medical files of patients and collected the relevant data. Data on demographics, signs, and symptoms from admission until hospital discharge were obtained.

All patients (or their legal representatives) signed an informed consent form. This study was approved by the Ethical Committee of University Hospital Clementino Fraga Filho (HUCFF) [number CAAE:31240120.0.0000.5257]. After matching for comorbidities, sex, and age group, 15 patients were selected for the analysis.

### 4.2. Herpesvirus Detection

Viral DNA was extracted from nasal swab samples using a viral DNA extraction kit (Qiagen, Valencia, CA, USA) according to the manufacturer’s recommendations. The viral loads of alphaherpesviruses (HSV-1, HSV-2, and VZV), betaherpesviruses (CMV, HHV-6, and HHV-7), and gammaherpesviruses(EBV and HHV-8) were quantified using real-time TaqMan PCR, as described previously by our group [18].

### 4.3. miRNAs Associated with COVID-19 and Neuroinflammation

Based on an extensive literature search, it was selected miRNAs associated with neurological disease/inflammation (miR-21, miR-146a, miR-let-7b, and miR-155-5p) [30,31,33,34,35,36,37]. We evaluate each miRNA expression in three COVID-19 patient groups: Group 1, HHV-6 detection, and neurological disease diagnosis (*n* = 5); Group 2, without HHVs detection and neurological disease (*n* = 5); and Group 3, without both HHVs and neurological disease (*n* = 5). We analyzed the expression profiles of these miRNAs in 15 serum samples from patients with COVID-19, in whom the detection of HHV-6 had been previously investigated by our group [18] (Figure 2).

### 4.4. miRNA Isolation

The selected miRNAs were validated using TaqMan miRNA assays. Total RNA was extracted from 250 µL of serum samples using from serum samples using the mirVana™ PARIS™ Kit (Life Technologies, Carlsbad, CA, USA). Five microliters of synthetic cel-miR-39-3pUCACCGGGUGUAAAUCAGCUUG (1 fmol/µL) (Life Technologies, Carlsbad, CA, USA) and cel-miR-54-5p AGGAUAUGAGACGACGAGAACA (1 fmol/µL) (Life Technologies, Carlsbad, CA, USA) were spiked-in before RNA isolation for normalization of the subsequent quantitative real-time PCR (qRT-PCR). Then, the extraction step occurred according to the manufacturer’s protocol and the RNA was eluted from the columns with 60 μL of eluent solution provided by the kit.

### 4.5. cDNA Synthesis, and Quantification of miRNAs by qPCR

The isolated RNA was reverse transcribed to complementary DNA (cDNA) using the TaqMan MicroRNA Reverse Transcription Kit (Applied Biosystems, Beverly, MA, EUA) and specific miRNA RT-primers for each reaction (total reaction volume, 15 μL), following the manufacturer’s guidelines. RT-qPCR was performed using TaqMan™ Universal Master Mix II with UNG (ThermoFisher Scientific, Waltham, MA, USA) and specific TaqMan miR assays (Applied Biosystems^®^) for hsa-miR-146a-5p (assay ID 000468), hsa-miR155 (assay ID002623), hsa-let-7b-5p (assay ID 002619), hsa-miR-21-5p (assay ID 000397), hsa-miR-124-3p (assay ID 003188), cel-miR-54-5p (assay ID 462123_mat), and cel-miR-39-3p (assay ID 000200) on the QuantStudio 3 system (Thermo Fisher Scientific, Waltham, MA, USA). The amplification conditions were as follows: 2 min at 50 °C and 10 min at 95 °C, followed by 40 cycles of 15 s at 95 °C and 1 min at 60 °C. Data were analyzed using the Expression Suite version 1.3 program (Thermo Fisher Scientific, Waltham, MA, USA). The threshold was set at 0.02 for all targets. Relative quantification was performed using the 2^∆∆Ct^ method (ΔCq = Cq_miRNA_ − Cq_cel-miR-39-3p_) [64,65], to obtain the Fold Change (FC) values.

### 4.6. Statistical Analysis

The Shapiro-Wilk test was performed to determine normality, miR-21 (*p* = 0.654), miR-155 (*p* = 0.436), miR-146a (0.885) and miR-let-7b (0.285). For the equal variance test was used for analysis of variance, getting the following values miR-21 (*p* = 0.050), miR-155 (*p* = 0.301), miR-146a (0.050) and miR-let-7b (0.405). To identify significant differences between groups, using the Cq values, one-way ANOVA (miR-155 and miR-let-7b) or Kruskal-Wallis test (miR-21 and miR-146a) was applied. For comparisons between different groups, we used t test or Mann-Whitney test, according to parametric or non-parametric description between groups. Statistical significance was set at *p* < 0.05. Statistical analyses were performed using Sigma Plot 12.0 (Chicago, IL, USA). For graphics, GraphPad Prism 9 (La Jolla, CA, USA) was employed.

## Figures and Tables

**Figure 1 ijms-24-11201-f001:**
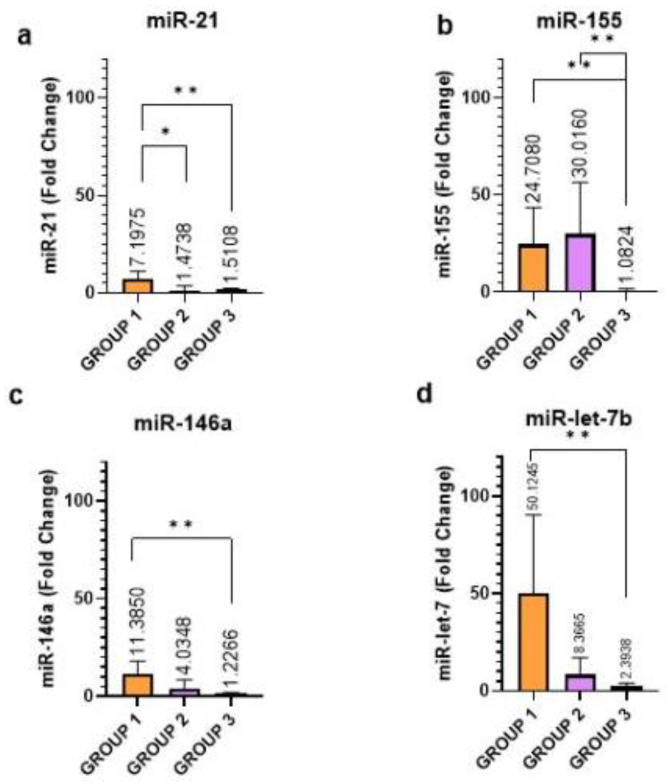
Relative expression of miR-21 (**a**), miR-155 (**b**), mir-146a (**c**), and miR-let-7b (**d**) in 3 different patient groups. *p* values indicate the significance level for each comparison based on ANOVA or Kruskal-Wallis test. The average FC of each group is indicated in the graph. In the multiple comparison analysis, the t test or Mann-Whitney test was used. ** *p* < 0.01 and * *p* < 0.05.

**Figure 2 ijms-24-11201-f002:**
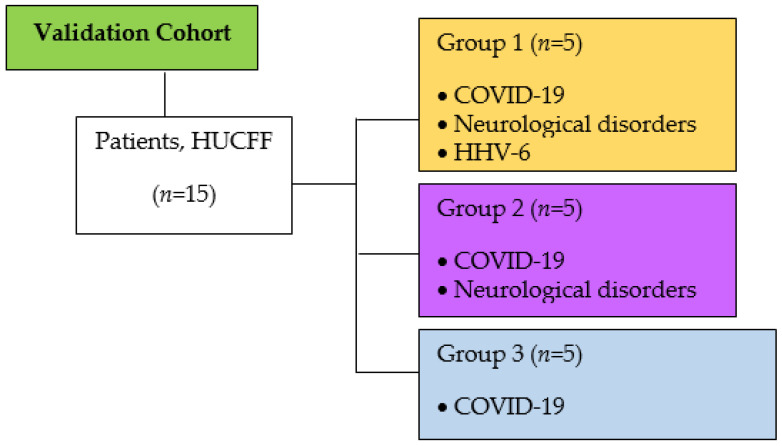
Schematic representation of the set of patient samples used to validate the miRNAs associated with neuroinflammation. Patients, HUCFF [18].

**Table 1 ijms-24-11201-t001:** Characteristics of the studied population.

Characteristics of the Studied Population			
	Group 1 (*n* = 5)	Group 2 ^#^ (*n* = 4)	Group 3 (*n* = 5)
Demographic characteristics			
Age	69.8 (±9.497)	71.25 (±11.53)	59.80 (±12.40)
Sex			
Female	3 (60.0)	3 (75.0)	1 (20.0)
Male	2 (40.0)	1 (25.0)	4 (80.0)
Clinical characteristics			
Hypertension	4 (80.0)	4 (100)	2 (40.0)
Diabetes	2 (40.0)	1 (25)	2 (40.0)
Pre-existing heart or cerebrovascular disease	3 (60.0)	0	0
SOFA score > 9	1 (20.0)	3 (60.0)	1 (20.0)
Death	1 (20.0)	3 (60.0)	1 (20.0)

Group 1 patients with COVID-19, neurological disorder, and HHV-6; Group 2 patients with COVID-19, neurological disorder, and without HHVs; Group 3 patients with COVID-19. ^#^ One sample from Group 2 was excluded as the miRNA quantification result of this sample did not pass the quality control (high variability in spike-ins). SOFA score: ≤ 9 (mild to moderate) >  9 (severe).

**Table 2 ijms-24-11201-t002:** Frequency of neurological symptoms among COVID-19 patients.

Symptoms	Group 1 (*n* = 5)	Group 2 (*n* = 4)
**Central Nervous System**		
Impaired Consciousness	3 (60.0)	1 (25.0)
Headache	2 (50.0)	2 (40.0)
Acute Cerebrovascular Disease	1 (20.0)	0
**Peripheral Nervous System**		
Hypo/Ageusia	0	2 (50.0)
Hypo/Anosmia	0	1 (25.0)
Vision impairment	0	1 (25.0)

Group 1 patients with COVID-19, neurological disorders, and HHV-6; Group 2 patients with COVID-19, neurological disorders, and without HHVs.

## Data Availability

Not applicable.

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
