# Peer review of "miRNAs in Neurological Manifestation in Patients Co-Infected with SARS-CoV-2 and Herpesvírus 6 (HHV-6)"

_ijms, 2023, doi:10.3390/ijms241311201_

Round 1
Reviewer 1 Report
The authors describe in this original article a relationship HHV-6 detection and neurological symptom in COVID-19 patients and possibility of miRNA as biomarker.
The manuscript is easy to understand authors’ enforcement, but some clarifications about describing the study design and interpretation of results are needed.
Comments:
1. I think that this study presupposed authors’ previous study (ref 6). It is difficult to interpret this study without understanding incidence rate of HHV-6 positive in COVID-19 patients with neurological symptom and so on. Detecting CMV, HHV-6 and HHV-7 from nasal swab is also not common way. I think describing about the previous study in introduction makes it clearer.
2. Abstract, Page1, line28. HHV-6 DNA?
3. Abstract, Page1, Line29.Which group does “this group” indicate?
4. Page3, Table1. Please describe about SOFA score in figure legend or method.
5. Page3, line122, Table2. This table indicates neurological symptom between Group 1 and 2, and high incidence of central nervous system disorder in HHV-6+ group. This study doesn’t describe total number of COVID-19 in this hospital, incidence of neurological symptom in them and incidence of HHV-6 in patient with neurological symptom. Therefore, interpretating this table may raise concerns bias of patient selection in each group (including arbitrary selection). Does describing your previous study (ref 6) resolve it?
6. This study doesn’t include the group; HHV-6(+) and neurological symptom (‐) for as negative control. Therefore, for miR-21, 146a and let‐7, which did not show significant difference between group2 and 3, it can’t deny the possibility that HHV-6 reactivation was just the main cause of these marker elevations, even without neurological symptom. Especially, miR-21 which is biomarker of neurological disease was lower in group2 than in group3. Please describe these points. I think low incidence of HHV-6 detection in patients without neurological symptom may help resolving this problem.
7. Page7, line271. Please describe timing of collection samples. If samples were collected prior to the onset of neurological symptoms, it supports that HHV-6 detection and elevated miRNA levels cause neurological symptoms.
8. Page7, line291. “i-155-5p” means miR-155-5p?
9. Page7, line294. Group2 and 3; without HHV-6 or without HHVs?
10. Page9、line328. I understand that the authors confirmed normality in miR-155 and let-7, then ANOVA and T test were performed in these analyses. Describing the result of the Shapiro-Wilk test may be better.

Author Response
Response to Reviewer 1 Comments
Point 1- I think that this study presupposed authors’ previous study (ref 6). It is difficult to interpret this study without understanding incidence rate of HHV-6 positive in COVID-19 patients with neurological symptom and so on. Detecting CMV, HHV-6 and HHV-7 from nasal swab is also not common way. I think describing about the previous study in introduction makes it clearer.
Response 1: As suggested by reviewer we described more information about our previous study findings (ref.6) (lines 47-57):
“Recently, we analyzed nasopharyngeal swab samples collected from hospitalized patients with COVID-19 diagnosis confirmed, to investigate the frequency of the detection of all types of herpesviruses in critically ill patients with COVID-19, and thus, to evaluate if SARS-CoV-2 infection could be a risk factor for Herpesviridae reactivation. It was observed that CNS symptoms were more prevalent in patients with HHV-6 detection (40% in patients with HHV-6 detection vs. 14.3% in patients without HHV-6 detection), showing a statistically significant association between HHV-6 detection and neurological symptoms in patients with COVID-19 [10].” The detection of CMV, HHV-6 and HHV-7 from nasal swab has been used to investigate Herpesviridae reactivation in previous studies [11,12], since the detection of herpes close to the sites of latency may indicate reactivation (line 47-56)
Point 2- Abstract, Page1, line28. HHV-6 DNA?
Response 2: In this sentence we refer to a group of patients without detection of HHVs-DNA (line 28). This group includes samples that did not detect DNA from any HHVs, as well as HHV-6. We were careful to select samples without detecting any herpesviruses to avoid a confounding factor.
Point 3-Abstract, Page1, Line29.Which group does “this group” indicate?
Response 3: We appreciate the observation. In this sentence we indicated the group with detection of HHV-6 and neurological alteration (line 29).
Point 4- Page3, Table1. Please describe about SOFA score in figure legend or method.
Response 4: Thank you for pointing this out. The SOFA score was described in the figure caption (line 130)
Point 5- Page3, line122, Table2. This table indicates neurological symptom between Group 1 and 2, and high incidence of central nervous system disorder in HHV-6+ group. This study doesn’t describe total number of COVID-19 in this hospital, incidence of neurological symptom in them and incidence of HHV-6 in patient with neurological symptom. Therefore, interpretating this table may raise concerns bias of patient selection in each group (including arbitrary selection). Does describing your previous study (ref 6) resolve it?
Response 5: As suggested by reviewer, we added more information about our previous study:
In our previous study, patients with mild or severe COVID-19 were further categorized according to their HHV-6 infection status and neurological diseases. 26.4% of them had neurological symptoms associated with the central nervous system, and among them 40% had HHV-6 DNA detected [10]. For the present study, it was selected 15 patients of these cohort, after matching for comorbidities, sex, and age group. (Line 306-311).
We also discussed better this point at the discussion: “In this study, we were the concerned about distinguishing the groups, avoiding confounding factors, which is common in studies that evaluate the expression of miRNAs. Thus, in group 2, it was included patients who did not detect any HHVs. Furthermore, the selection of samples was made carefully evaluating several intrinsic factors, such as homogeneity of pair samples between groups, considering age and detection of HHVs” (lines 268-272).
Point 6-This study doesn’t include the group; HHV-6(+) and neurological symptom (‐) for as negative control. Therefore, for miR-21, 146a and let‐7, which did not show significant difference between group2 and 3, it can’t deny the possibility that HHV-6 reactivation was just the main cause of these marker elevations, even without neurological symptom. Especially, miR-21 which is biomarker of neurological disease was lower in group2 than in group3. Please describe these points. I think low incidence of HHV-6 detection in patients without neurological symptom may help resolving this problem.
Response 6: To clarify we discussed better these points: “We must consider that the proposed objective of this study is not to associate these miRNAs with neurological symptoms, this association is already seen in the literature, as described. However, we sought to associate the expression of these miRNAs with neurological symptoms associated with the detection of HHV-6, which was observed in group 1. This is new and highly relevant data, as studies associating these miRNAs with neurological symptoms already exist, but studies linking this association to HHV-6 have not been described. Although it is unclear whether the increased expression of miR-21, miR-146a and miR-let-7b in group 1 is related to neurological symptoms or HHV-6 detection only, given the low expression of these miRNAs in group 2, without detection HHV-6, but with neurological symptoms. Despite in this study it was not included a control group of HHV-6 (+) and neurological symptom (‐), due to the low incidence of HHV-6 detection in patients without neurological symptom (14.3%) [10], we observed that miR-21, 146a and let‐7 did not show significant difference between groups 2 and 3, which strongly suggest that HHV-6 reactivation was the main cause of these miRNAs elevations, even without neurological symptom, indicating that HHV-6 infection may play a role in the expression of these miRNAs, corroborating neurological manifestations during COVID-19. The expression level of all miRNAs was more elevated among group 2, compared with group 3, except for miRNA 21, that showed a slightly increase among group 3. However, these differences were not statistically different. These findings reinforce the role of these miRNAs in neurological symptoms. (line 226-245).
Point 7-Page7, line271. Please describe timing of collection samples. If samples were collected prior to the onset of neurological symptoms, it supports that HHV-6 detection and elevated miRNA levels cause neurological symptoms.
Response 7: Samples were collected after the onset of respiratory symptoms, and any change in the neurological status of patients after collection was observed and evaluated, by neurologists who collaborated in the study. However, the time between the onset of respiratory symptoms and sample collection was short but varied among patients.
We included these details in the Materials and Methods of the manuscript (line 311-314)
Point 8-Page7, line291. “i-155-5p” means miR-155-5p?
Response 8: Thanks for pointing this out. It was corrected to miR-155-5p (line 331)
Point 9-Page7, line294. Group2 and 3; without HHV-6 or without HHVs?
Response 9: We appreciate the note. The text has been corrected to “without HHVs” (line 333-334).
Point 10- Page9、line328. I understand that the authors confirmed normality in miR-155 and let-7, then ANOVA and T test were performed in these analyses. Describing the result of the Shapiro-Wilk test may be better.
Response 10: As suggested by reviewer, this information was included in the manuscript (line 367-368)
The Shapiro-Wilk test was performed to determine normality in miR-21 (p=0.654), miR-155 (p=0.436), miR-146a (0.885) and miR-let-7b (0.285).
Reviewer 2 Report
Title
miRNAs in neurological manifestation in patients co-infected with SARS-CoV-2 and Herpesvirus 6 (HHV-6)
Summary
SARS-CoV-2/HHV-6 co-infected individuals have neurological symptoms. The authors report consistently increased neuroinflammatory biomarker miRNAs in SARS-CoV-2/HHV-6 co-infected individuals relative to SARS-CoV-2 infected controls with and without neurological symptoms. Levels of individual biomarkers are shown and described. These biomarkers are not specific to this condition, but have potential for clinical value in the context of other metrics, patient history and communication.
Comments
As an exploratory description of miRNAs from a highly-specific population (coinfected individuals), the sample size is small but understandably so. There are a few punctuation and grammatical errors that should be remedied in copy-edit, but otherwise this observation should be shared and is appropriate for this journal.
Line 70: perhaps an ideal biomarker would also be minimally-invasive to collect.
There are a few punctuation and grammatical errors that should be remedied in copy-edit, but otherwise this observation should be shared and is appropriate for this journal.
Author Response
Response to Reviewer 2 Comments
As an exploratory description of miRNAs from a highly-specific population (coinfected individuals), the sample size is small but understandably so. There are a few punctuation and grammatical errors that should be remedied in copy-edit, but otherwise this observation should be shared and is appropriate for this journal.
Point 1- Line 70: perhaps an ideal biomarker would also be minimally-invasive to collect.
Response 1: We have added this information at this line (line 79)
Point 2- There are a few punctuation and grammatical errors that should be remedied in copy-edit, but otherwise this observation should be shared and is appropriate for this journal.
Response 2: We appreciate your comments and observations. The text has been revised and corrected, as indicated.
Round 2
Reviewer 1 Report
This manuscript was revised well according to the comments. However, some content are difficult to understand. Especially, the part of modified according to the comments does not seem to match the content of other discussion. I think it needs to be corrected further.
Please fing word file.

Author Response
Response to Reviewer 1 Comments
This manuscript was revised well according to the comments. However, some contents are difficult to understand. Especially, the part of modified according to the comments does not seem to match the content of other discussion. I think it needs to be corrected further. Comments:
Point 1. Page2, line47-56. I think it is better to transfer these sentences you added after line66 “or to MOGAD disease [16-18]” or after line68 “in specific individuals”.
Response 1: Thanks for the suggestion. These sentences were transfered to lines 59 to 68 (Line 59-68)
Point 2. Page6, line200-204. These two sentences mention similar contents repeatedly. Please put together.
Response 2: Thanks for pointing this out. The sentence has been changed:
“In the present study, the expression of miR-146a was observed to significantly differ among the patient group 1 (patients with COVID-19, neurological alterations, and detection of HHV-6) and group 3 (patients with COVID-19). miR-146a is reported to be one of the main miRNAs that regulate neuroinflammation, usually with a suppressive effect [54]. Studies have linked the differential expression of mir-146a to neurological disorders, such as Alzheimer's disease [55]. (line 201-206)”
Point 3. Page7, Line226-245. This paragraph is too complicated, so that it is difficult to understand what you want to describe. These sentences seem to indicate against authors’ contention. I interpreted it as follows. does it match your contention? (I’m not native English, please correct it) “In this study, we sought to investigate the association between the expression of these miRNAs and the detection of HHV-6 in COVID-19 patients with neurological symptoms, which was observed in group 1. Due to the low incidence of HHV-6 detection in patients without neurological symptom (14.3%) [10], it was not included a control group of neurological symptoms (‐) and HHV-6 (+) in this study. Therefore, it is unclear whether the increased expression of miR-21, miR146a and miR-let-7b in group 1 is related to neurological symptoms with HHV-6 (+) or HHV-6 (+) only. However, we observed that miR-21, 146a and let‐7 did not show significant difference between groups 2 and 3. Moreover, CNS disorders tended to be higher in group 1 than group 2. As described above, the association of these miRNA with CNS disorders has already been demonstrated in several literatures. Given low incidence of HHV-6 detection in COVID-19 patients without neurological symptom in our previous study, it strongly suggests that HHV-6 reactivation induces CNS disorders and was the main cause of these miRNAs elevations, corroborating neurological manifestations during COVID-19. The expression level of all miRNAs was more elevated among group 2, compared with group 3, except for miRNA 21, that showed a slightly increase among group 3. However, these differences were not statistically different. These findings reinforce the role of these miRNAs in neurological symptoms. “
Response 3: Thank you for your comment and your interpretation is correct. To facilitate the reader's understanding, we chose to include his interpretation of the text, with just a few changes.
“In this study, we sought to investigate association between the expression of these miRNAs and the detection of HHV-6 in COVID-19 patients with neurological symptoms, which was observed in group 1. Due to the low incidence of HHV-6 detection in patients without neurological symptom (14.3%) [10], it was not included a control group of neurological symptoms (‐) and HHV-6 (+) in this study. Therefore, it is unclear whether the increased expression of miR-21, miR146a and miR-let-7b in group 1 is related to neurological symptoms with HHV-6 (+) or HHV-6 only (+). However, we observed that miR-21, 146a and let‐7 did not show significant difference between groups 2 and 3. Moreover, CNS disorders tended to be higher in group 1 than group 2. As described above, the association of these miRNA with CNS disorders has already been demonstrated in literatures. The low incidence of HHV-6 detection in COVID-19 patients without neurological symptom in our previous study, it strongly suggests that HHV-6 reactivation induces CNS disorders and was the main cause of these miRNAs elevations, corroborating neurological manifestations during COVID-19. The expression level of all miRNAs was more elevated among group 2, compared with group 3, except for miRNA 21, that showed a slightly increase among group 3. However, these differences were not statistically different. These findings reinforce the role of these miRNAs in neurological symptoms. “ (line 225-242)
Point 4. Page10, line367. Are these p-values by Shapiro-wilk test correct? Could you confirm it? All of these results are p>0.05, so that I think they can be judged to follow the normal distribution.
Response 4: Although all passed the Shapiro Wilk test, statistical evaluation of differences was performed with one-way ANOVA or the Kruskal Wallis test, depending on the results of normality and tests of equal variance. When the equal variance test failed, a non-parametric test was used. Results of the equal variance test have been included in the text.
The Shapiro-Wilk test was performed to determine normality, miR-21 (p=0.654), miR-155 (p=0.436), miR-146a (0.885) and miR-let-7b (0.285). For the test of equal variance, analysis of variance was used, obtaining the following values miR-21 (p=0.050), miR-155 (p=0.301), miR-146a (0.050) and miR-let-7b (0.405 ) . (line 364-367)
.